# Risk Factors for Long-Term Contracture Recurrence after Collagenase Injection for Dupuytren Disease: A Prospective Cohort Study

**DOI:** 10.3390/biomedicines11030699

**Published:** 2023-02-24

**Authors:** David Eckerdal, Anna Lauritzson, Anna Åkesson, Isam Atroshi

**Affiliations:** 1Department of Orthopedics, Hässleholm-Kristianstad Hospitals, 28 136 Hässleholm, Sweden; 2Department of Clinical Sciences—Orthopedics, Lund University, 223 62 Lund, Sweden; 3Department of Rehabilitation, Hässleholm-Kristianstad Hospitals, 28 125 Hässleholm, Sweden; 4Clinical Studies Sweden—Forum South, Skåne University Hospital, 221 85 Lund, Sweden

**Keywords:** Dupuytren, contracture, collagenase, recurrence

## Abstract

In Dupuytren disease, little is known about the long-term outcomes of collagenase injection or risk factors for contracture recurrence. In this prospective study, 159 patients (242 fingers) with Dupuytren disease and active extension deficit (AED) ≥20° in a metacarpophalangeal (MCP) or proximal interphalangeal (PIP) joint were treated with collagenase injection during a 14-month period. At 5 years, 18 patients were deceased, 2 could not be contacted, and 13 had undergone fasciectomy. The remaining 126 patients (199 treated fingers) participated in a follow-up evaluation at 52–96 (mean 65) months after injection, with physical examination (114 patients) or telephone interview (12 patients). Recurrence was defined as subsequent treatment (surgery or repeat injection) or ≥20° AED worsening in a treated joint between the 6-week and 5-year measurements. The mean AED at baseline was 42° (SD 24) for MCP joints and 31° (SD 29) for PIP joints and at 5 years 11° (SD 17) and 17° (SD 23), respectively. Recurrence occurred in 17% of MCP joints and 25% of PIP joints. Statistically significant risk factors for PIP joint contracture recurrence were greater baseline AED (odds ratio 1.04, 95% CI 1.02–1.06) and small finger treatment (OR 4.6, 95% CI 1.5–14.3), with no significant risk factors found for MCP contracture recurrence.

## 1. Introduction

Regardless of treatment modality, recurrences are common after treatment for Dupuytren disease [1,2,3]. The Dupuytren diathesis includes male sex, bilateral disease, north European ethnicity, ectopic disease, onset of disease before the age of 50 years, and family history [4]. The Dupuytren diathesis has been shown to increase an individual’s predisposition for Dupuytren disease and predict a more aggressive disease. Smoking and diabetes have been reported to increase the risk of developing Dupuytren disease [5] but, while Dupuytren diathesis is known to increase the risk of recurrence [4], the role of diabetes and smoking is still unclear.

After its introduction, collagenase is today considered a standard treatment for Dupuytren disease together with surgical fasciectomy and needle fasciotomy. Citing commercial reasons, the manufacturer withdrew the marketing authorization in all countries except for the United States in early 2020. Consequently, collagenase injection for Dupuytren disease is currently unavailable outside the United States. However, collagenase is still frequently used in the United States. According to a recent study of treatment trends in the United States, collagenase injection was used in almost 25% of the patients that were treated for Dupuytren disease between October 2015 and September 2019 [6].

Despite that more than 10 years has passed since the initial study by Hurst et al. [7], data about long-term outcomes of collagenase injections for Dupuytren disease are still scarce. Although 5-year results of the initial multicenter “Cordless” study have been reported [2], these were based on the original collagenase treatment method. The method has since been modified and local anesthesia before extension as well as treatment of multiple joints and fingers using a higher collagenase dose is now standard [8,9,10].

Few prospective studies have reported long-term results using this modified technique. In addition, little is known about the factors that predict long-term contracture recurrence after treatment with collagenase injection. In a previous study comparing 5-year outcomes after collagenase and surgical fasciectomy, we have shown a similar prevalence of joint contractures in the treated fingers after both treatments [11]. Using the same cohort, the purpose of this study is to evaluate the 5-year outcomes of collagenase injections and determine the risk factors for recurrence. We hypothesized that in patients with Dupuytren disease that were treated with collagenase injection, the long-term recurrence rate would be high.

## 2. Materials and Methods

This prospective single-center cohort study was conducted at an orthopedic university department in southern Sweden. At the time of the study, this was the only department that treated patients for Dupuytren disease in a region with 300,000 inhabitants.

The study was approved by the Regional Ethical Review Board in Lund (2013/656, 10-17-2013). The inclusion criteria were Dupuytren disease involving one or more of the three ulnar fingers, palpable cord, and a joint contracture of ≥20° in the metacarpophalangeal (MCP) and/or the proximal interphalangeal (PIP) joint. All patients that were treated with collagenase injection between September 2013 to October 2014 were eligible for participation in a 5-year follow-up evaluation. Patients who had subsequent surgical fasciectomy after the injection were considered to have had recurrence and thus not asked to attend examination.

### 2.1. Collagenase Injection and Finger Manipulation

After local anesthesia (administered as a nerve block using 10 mL of 10 mg/mL mepivacaine), a hand surgeon injected 0.8 mg of collagenase (Xiapex, Endo pharmaceuticals, Malvern, PA, USA) distributed over multiple locations of the Dupuytren cord. The patients received a soft dressing and were given instructions by an occupational hand therapist. The patients returned 1 or 2 days after collagenase injection and after a similar nerve block, the treating surgeon performed manipulation of the finger to best achievable extension. When skin tears occurred, these were treated with dressings until healed [12]. All patients received a static night splint to be used for 2 months. The patients visited the occupational hand therapist 1 week after injection for measurement of extension deficit as well as adjustment of the night splint if required. Additional visits to the occupational hand therapist were planned when necessary.

### 2.2. Measurements

At baseline, active extension deficit (AED) was recorded by an occupational hand therapist immediately before collagenase injection. At the 5-year follow-up examinations that were conducted between November 2018 and March 2020, AED was measured by either an occupational hand therapist or orthopedic resident. All measurements were performed using a 5° increments standard 6-inch metal goniometer (Baseline, Fabrication Enterprises, White Plains, NY, USA). All patients completed the 11-item Disabilities of the Arm, Shoulder, and Hand (QuickDASH) scale and rated their treatment satisfaction for each treated hand on a 5-point scale (1 = very satisfied, 2 = satisfied, 3 = rather satisfied, 4 = neutral, 5 = dissatisfied).

### 2.3. Statistical Analysis

We calculated the mean and SD for AED at baseline, 6 weeks, and 5 years (hyperextension was recorded as 0°). We calculated the mean differences (baseline to 6 weeks and 6 weeks to 5 years) with 95% confidence intervals (CI) and analyzed them with the paired *t*-test. The primary outcome was recurrence which was defined as AED worsening by ≥20° between the 6-week and the 5-year follow-up, subsequent surgery, or repeat injection. For joints with a baseline contracture of ≥20°, we calculated the proportion with complete correction, defined as AED ≤5°. We used a mixed effects regression model to determine the risk factors for recurrence and odds ratios (OR) and 95% CI were calculated for both MCP and PIP joints. An additional analysis was performed with the same model in which repeat injections were not considered as recurrence. In both models, the covariates (age, sex, baseline contracture, small-finger treatment, previous treatment with surgery or injection, diabetes and smoking) were chosen based on the risk factors that were included in the Dupuytren diathesis and those that were reported in previous studies of recurrence after collagenase injection [4,13]. We calculated the mean and SD for the QuickDASH scores and the proportions for each patient-reported satisfaction level; patients who had undergone subsequent surgery were considered as dissatisfied.

A *p*-value <0.05 was used for statistical significance. All statistical analyses were performed using Stata SE Release 16 (StataCorp LLC, College Station, TX, USA) and SPSS version 25 (IBM Corp, Armonk, NY, USA).

## 3. Results

### 3.1. Patients

Of 159 consecutive patients (178 treated hands, 242 treated fingers) who fulfilled the study’s eligibility criteria, 18 patients were deceased, 2 patients could not be reached, and 13 patients (17 fingers) had received subsequent surgical fasciectomy (Figure 1).

Clinical examination was conducted on 114 patients (182 treated fingers) and an additional 12 patients (17 treated fingers) participated in a telephone interview about their hand status. The mean time from injection to follow-up was 65 months (SD 9, range 54–96). Of the study participants, 89% were men, the small finger constituted almost half of the treated fingers, and 8% had undergone previous fasciectomy (Table 1). Prior to the 5-year follow-up, repeat injections had been given to 17 patients (18 hands, 23 fingers) because of recurrent or worsened contracture.

### 3.2. Change in Joint Contracture

For all the treated fingers, the mean (SD) AED at baseline was 42° (24) for MCP joints and 31° (29) for PIP joints and the mean TAED was 72° (36). The corresponding values at the 6-week follow-up were 9° (12), 11° (15), and 20° (21) and at the 5-year follow-up 11° (17), 17° (23), and 28° (31) (Table 2).

Of the joints with a baseline AED ≥20° (176 MCP and 127 PIP), and when considering joints that had received repeat injection as not having complete correction irrespective of status at 5 years, complete correction was observed in 88 (50%) of MCP joints and 34 (27%) of PIP joints. Based on 5-year status including joints with previous repeat injection, complete correction at 5 years was present in 95 MCP joints (54%) and 38 PIP joints (30%).

### 3.3. Risk Factors for Contracture Recurrence

Recurrence was observed in 37 MCP joints (17%) and 53 PIP joints (25%) and in either of the joints in 79 (36%) of the treated fingers. In the first model (both subsequent surgery and subsequent injection considered as recurrence), statistically significant risk factors for PIP contracture recurrence were greater baseline AED and treatment of the small finger (Table 3). For baseline AED, the OR (per degree) was 1.04 (95% CI 1.02–1.06) and for treatment of small-finger PIP joint (compared with long and ring finger), the OR was 4.6 (95% CI 1.5–14.3). Age, sex, previous injection, diabetes, and smoking were not significant risk factors for recurrence. 

In the second model (having received repeat injection not considered as recurrence by default but by actual 5-year status), the same risk factors for recurrence were found (Table 4). For baseline contracture, the OR was 1.04 (95% CI 1.02–1.06), and for small-finger treatment the OR was 3.98 (95% CI 1.31–12.04).

For MCP joint recurrence, no statistically significant risk factors were found in either of the two models.

### 3.4. Patient-Reported Outcomes

The mean QuickDASH score (available for 107 hands [74%]) was 10 (SD 12). Self-rated satisfaction with the treatment outcome was available for all 145 hands that were included in the 5-year follow-up. Patients reported being very satisfied in 77 hands (49%), satisfied in 28 hands (18%), rather satisfied in 16 hands (10%), neutral in 12 hands (8%), and dissatisfied in 12 hands, and with the 13 hands that underwent subsequent surgery added, the dissatisfied group was comprised of 25 hands (16%).

### 3.5. Adverse Events

In the entire cohort, skin tears after finger manipulation occurred in 62 hands (35%). One patient developed pain and swelling after injection, which was diagnosed as a complex regional pain syndrome and resolved within 4 months following hand therapy.

## 4. Discussion

In this study assessing the long-term outcomes after collagenase injection for Dupuytren disease with a mean follow up of 65 months and minimum of 54 months, recurrence (defined as AED worsening by ≥20° between the 6-week and the 5-year follow-up, subsequent surgery, or repeat injection) was observed in 17% and 25% of MCP and PIP joints, respectively. Despite the use of AED instead of the more common passive extension deficit (PED), the results are better than in many previous studies. A randomized controlled study that compared collagenase with needle fasciotomy, but only addressed MCP joints, reported a recurrence rate of 56% at 5 years [14]. Another randomized study of mainly PIP joints reported recurrences in 83% of PIP joints at 2 years [15]. In both these studies, previous treatment was an exclusion criterium. However, we believe that it is important to include all patients and all joints that were treated, both those with primary treatment as well as those that were previously treated to reflect clinical practice. Also, since previous fasciectomy has been reported as a risk factor for recurrence, inclusion of patients with previous treatment further strengthens the results of present study [13]. It is also important to consider that the definition of recurrence varies between studies. In contrast to the present study in which recurrence was defined as worsening of ≥20° from the early follow-up, both studies that were mentioned above defined recurrence as a joint contracture ≥20° at follow-up, thus introducing possible bias by including those that may have initially failed intervention [14,15]. In a recent study using the same cohort, current contracture (defined as a joint contracture ≥20°) was present in 25% of MCP and 33% of PIP joints [11]. Current joint contracture at follow-up does not necessarily mean recurrence because it could be disease extension or incomplete initial correction.

In the present study, we also performed analyses in which we considered repeat injections as recurrence irrespective of the 5-year joint status. It can be argued that, in contrast to surgical fasciectomy, collagenase injection is a simple minimally invasive treatment and, therefore, repeat injections should not be considered as failure but as part of the treatment method. In fact it has been suggested that recurrences should be considered part of the disease and not as failure [16]. However, as many patients may consider the need for repeat injections as treatment failure, thorough pre-treatment information about the risks and benefits of different treatment options is important to allow patients make informed decisions regarding treatment choices.

As there currently is no cure for Dupuytren disease, recurrence is expected. The Dupuytren diathesis has previously been established to increase the risk of recurrence after treatment for Dupuytren disease [4]. In this study, we showed that greater baseline contracture and treatment of the small finger were risk factors for recurrence in PIP joint contracture, while no risk factors were found for recurrence of contracture in the MCP joints. The PIP joints have previously been established to have higher rates of recurrence after treatment than MCP joints [2,13,17]. In our previous study with a 3-year follow-up of a separate cohort, treatment of the small finger as well as greater baseline contracture were also identified as risk factors for recurrence [13]. In the present study, these two factors were identified as risk factors for recurrence only for PIP contracture in contrast to the 3-year follow-up study where previous surgical fasciectomy was also found to be a risk factor [13]. This may be explained by the higher proportion of fingers with previous surgical fasciectomy in the 3-year study than in the present study (15/126 vs 19/242). The findings regarding risk factors for recurrence will be helpful when informing patients about expected treatment outcomes and discussing treatment choices.

At the 5-year follow-up, two-thirds of the patients rated their satisfaction with their treated hand as either very satisfied or satisfied, whereas 16% were dissatisfied or had undergone subsequent surgery. These results are generally consistent with previous studies using similar satisfaction ratings [13,18,19]. Also in a recent long-term (5 years) assessment of treatment satisfaction, the average satisfaction was 6.5 on a 1 to 10 scale, indicating a similar high satisfaction with treatment [20]. The mean QuickDASH of 10 in our study is higher than what is reported in other studies (mean 4.5, and median 4.5) [14,21] but these studies excluded patients with severe PIP contractures, a well-known risk factor for recurrence [13]. The mean score in our study is, however, comparable to those after surgical fasciectomy as presented in a recent study from our center [11].

There is a debate in the Dupuytren field regarding potential superiority of certain treatment methods, specifically collagenase injection versus needle fasciotomy. The results of the current study regarding recurrence are comparable to those of limited fasciectomy, where recurrence rates range from 12% to 39% after a mean follow-up of 1.5 to 7.3 years [22]. In a randomized study comparing needle fasciotomy with surgical fasciectomy Van Rijssen et al. reported recurrence rates (defined as worsening of passive total extension deficit by ≥30 between 6 weeks and 5 years) of 84.9% after needle fasciotomy. Also, the two studies mentioned above, that compared collagenase with needle fasciotomy, reported recurrence rates after needle fasciotomy of 68% and 45% at 2 and 5 years, respectively [14,15], which are substantially higher than those that were found in the present study. However, as mentioned above, these studies also had higher recurrence rates after collagenase injection than in the present study. We have previously shown in two other separate cohorts, similar recurrence rates of 14% for both MCP and PIP joint at 2 years and 14% and 23% for MCP and PIP joints, respectively, at 3 years [13,18]. There are many possible explanations for lower recurrence after collagenase compared to other studies. Differences in definition of recurrence may be a possible factor. The use of higher doses of collagenase injected into several locations in the Dupuytren cord may also contribute to lower rates of recurrence. Furthermore, in a previous study we found that skin tears after collagenase generally heal without complications [12]. Thus, when performing collagenase treatment, our goal during finger extension was to achieve the best possible contracture correction regardless of whether skin tears occurred. It is reasonable to believe that the use of local anesthesia as a nerve block would allow the treating surgeon to extend the finger further than without local anesthesia.

Most previous randomized studies have failed to show lower recurrence after collagenase compared to needle fasciotomy [14,23,24]. However, the substantially lower recurrence rates in the present study highlight the need for more research. Based on the similar results between collagenase and needle fasciotomy that were reported previously, it has been suggested that needle fasciotomy should be favored due to lower costs [23,25]. Our previous work as well as that of Zhao et al. have shown trends of fewer surgical fasciectomies after the introduction of collagenase, possibly suggesting that collagenase injections may be used as an alternative to surgical fasciectomy [13,26]. This, as well as the risks and potential costs of future surgeries for recurrent disease should be taken into account when considering cost effectiveness of treatments. Recurrent Dupuytren disease treated with surgical fasciectomy is associated with higher risks than primary surgery. Collagenase can also be used for treating recurrent disease, with good results shown at 1 year [27]. With regard to treatment of recurrent Dupuytren disease, evidence from RCTs is needed [28].

The main limitation of the present study is that no examinations were performed between the 6-week and the 5-year assessments, which would have enabled a more detailed description of change over time. We have taken this into account by considering all patients who had received subsequent treatments during the follow-up time as recurrences. Another limitation is the use of AED instead of the more commonly used PED. As AED is the range of motion the patient benefits from, it can be argued that AED might be more relevant from the patient perspective. Also, the results of our study are probably underestimated compared to those using PED. Another limitation is the lack of a parallel control group. Current treatment for Dupuytren disease includes three treatment options that are all regarded as effective and appropriately safe. Most of the current controversy concerns the relative merits of the different treatments and, therefore, studies of comparative effectiveness are important. We have however, as mentioned earlier, shown in a previous study similar rates of current contracture (defined as joint contracture ≥20°) in collagenase-treated joints and joints that were treated with surgical fasciectomy [11]. Still, as we also mention above, current contracture is not the same as recurrence.

Furthermore the lack of information regarding heredity, bilateral disease as well as ectopic disease, and age of disease onset, resulted in that these factors (all from the Dupuytren diathesis) known to be associated with recurrence [4], could not be included as covariates in our mixed models.

Finally, due to the lack of baseline QuickDASH and satisfaction ratings, these could not be included in the analysis of risk factors, but rather as patient-reported measures of long-term outcome. 

The strengths of the study include the long follow-up time with a mean of 65 months as well a high participation rate with long-term data available for 126 of 128 eligible patients, of whom 114 patients attended physical examination. Also, all treatments were provided by one surgeon, thus reducing the risk of differences in treatment affecting the outcome, although this may affect generalizability. Furthermore, as measurements vary between examiners, [29] measurements at the 5-year examination were performed by only two examiners using similar technique to minimize this effect. Another strength is that the assessment included measurements of joint contractures as well as patient-reported outcomes.

## Figures and Tables

**Figure 1 biomedicines-11-00699-f001:**
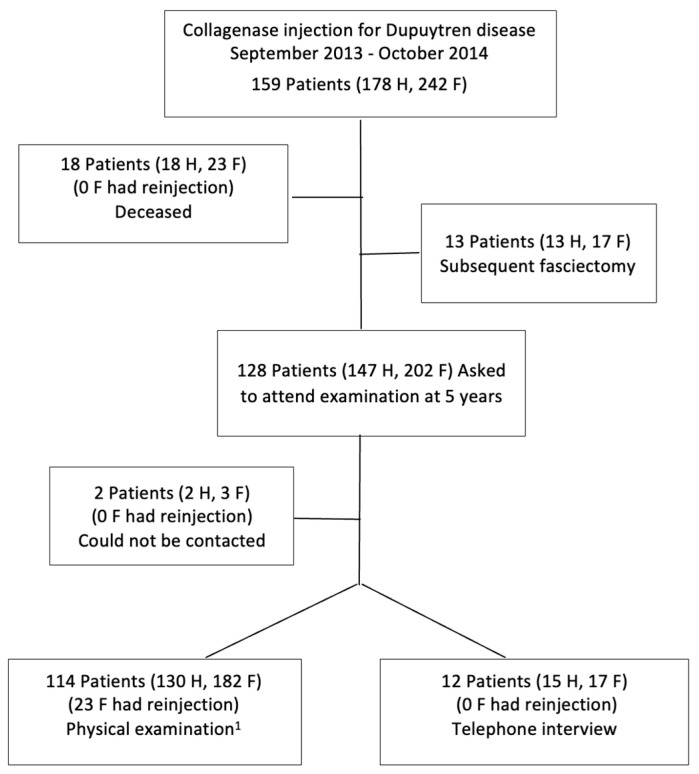
Flow chart of patients. ^1^ Follow-up time was 54–59 months for 8 patients. F, fingers; H, hands.

**Table 1 biomedicines-11-00699-t001:** Patient characteristics.

	Included	Subsequent Surgery	Deceased/Did Not Respond
Patients, *n*	126 ^1^	13	20
Age, mean (SD) y	69 (8)	60 (10)	72 (6)
Sex, *n* (%)			
Men	112 (89%)	11 (85%)	19 (95%)
Women	14 (11%)	2 (15%)	1 (6%)
Smoking/other tobacco, *n* (%)	17 (14%)	0 (0%)	0 (0%)
Diabetes, *n* (%)	12 (10%)	2 (15%)	5 (25%)
Treated hand, *n* (%)			
All	145	13	20
Right	83 (57%)	10 (77%)	8 (40%)
Left	62 (43%)	3 (23%)	12 (60%)
Treated finger, *n* (%)			
All	199	17	26
Small	95 (48%)	8 (47%)	13 (50%)
Ring	74 (37%)	7 (41%)	10 (38%)
Middle	30 (15%)	2 (12%)	3 (12%)
Previous treatment, *n* (%)			
Surgery	16 (8%)	1 (6%)	2 (8%)
Collagenase/NF	8 (4%)	1 (6%)	1 (4%)

NF, needle fasciotomy. ^1^ 12 patients, (15 hands, 17 fingers) follow-up via telephone interview.

**Table 2 biomedicines-11-00699-t002:** Active extension deficit in degrees.

				Mean Difference (95% CI)			
	Baseline *n* = 242	6 Weeks *n* = 242	5 Years *n* = 216	Baseline to 6 Weeks *n* = 242	*p*	6 Weeks to 5 Years *n* = 216	*p*
MCP	42 (24)	9 (12)	12 (18)	33 (30 to 36)	<0.001	−4 (−6 to −1)	0.002
PIP	31 (29)	11 (15)	19 (24)	20 (17 to 22)	<0.001	−8 (−10 to −5)	<0.001
TAED	72 (36)	20 (21)	31 (32)	53 (49 to 56)	<0.001	−11 (−15 to −8)	<0.001

CI, confidence interval; MCP, metacarpophalangeal; PIP, proximal interphalangeal; TAED, total active extension deficit (MCP + PIP).

**Table 3 biomedicines-11-00699-t003:** Risk factors of recurrence at 5 years; repeat injections were considered as recurrence.

	MCP		PIP	
Variable	OR (95% CI)	*p*	OR (95% CI)	*p*
Age (years)	0.97 (0.90–1.03)	0.321	0.98 (0.92–1.03)	0.384
Male sex	1.22 (0.22–6.90)	0.820	1.35 (0.31–6.01)	0.690
Baseline contracture (per°)	1.03 (0.999–1.05)	0.058	1.04 (1.02–1.06)	<0.001
Small finger (vs ring/middle)	2.35 (0.83–6.66)	0.107	4.56 (1.45–14.35)	0.010
Previous injection	1.63 (0.25–10.74)	0.611	1.68 (0.38–7.37)	0.494
Previous surgery	0.65 (0.04–11.04)	0.766	0.12 (0.01–2.93)	0.193
Diabetes	1.35 (0.24–7.46)	0.729	1.35 (0.30–6.10)	0.699
Smoking/other tobacco	1.60 (0.33–7.62)	0.558	0.80 (0.20–3.22)	0.748

CI, confidence interval; MCP, metacarpophalangeal; OR, odds ratio; PIP, proximal interphalangeal; TAED, total active extension deficit (MCP + PIP).

**Table 4 biomedicines-11-00699-t004:** Risk factors of recurrence at 5 years; repeat injections were not considered as recurrence.

	MCP		PIP	
Variable	OR (95% CI)	*p*	OR (95% CI)	*p*
Age (years)	0.96 (0.88–1.05)	0.347	0.98 (0.93–1.03)	0.447
Male sex	1.14 (0.12–11.11)	0.908	1.63 (0.40–6.57)	0.492
Baseline contracture (per°)	1.05 (1.01–1.08)	0.010	1.04 (1.02–1.06)	<0.001
Small finger (vs ring/middle)	0.39 (0.10–1.48)	0.166	3.98 (1.31–12.04)	0.015
Previous injection	1.63 (0.13–20.67)	0.705	0.89 (0.22–3.55)	0.869
Previous surgery	1.51 (0.06–39.49)	0.804	0.15 (0.01–3.01)	0.216
Diabetes	3.73 (0.43–32.25)	0.232	1.48 (0.36–6.06)	0.587
Smoking/other tobacco	3.9 (0.52–28.99)	0.184	0.71 (0.18–2.82)	0.629

CI, confidence interval; MCP, metacarpophalangeal; OR, odds ratio; PIP, proximal interphalangeal; TAED, total active extension deficit (MCP + PIP).

## Data Availability

The data will be made available to researchers upon reasonable request to the corresponding author and approval from the relevant authorities.

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
