# Peer review of "Risk Factors for Long-Term Contracture Recurrence after Collagenase Injection for Dupuytren Disease: A Prospective Cohort Study"

_biomedicines, 2023, doi:10.3390/biomedicines11030699_

Round 1

Reviewer 1 Report

This is a prospective cohort study of patients treated at a single center with collagenase for Dupuytrens Contracture (DD) of the MP & PIP joint. The investigation was conducted in Skane, Sweden and the senior author has published extensively on this topic. The inclusion criteria included a contracture of the small ring or long fingers of greater than or equal to 20 degrees at the MCP or PIP joint. In this manuscript the technique used is reasonably well described and differs slightly from the earliest publications describing treating DD contractures with collagenase. Contraction measurements were completed by a hand therapist or resident but not the treating surgeon. Recurrence was the primary outcome and this was defined by worsening of the extension deficit by greater than or equal to 20 degrees between the six weeks and five year follow-up time periods or subsequent treatment.

The rate of retreatment for recurrent contracture was 17% of fingers with an average time followed of 65 months (min 54months). Recurrence was seen in 17% of MCP joints, 25% of PIP joints and 36% of all fingers as some fingers had recurrence at both joints. The risk factors for recurrence identified corresponded to those previously published with the odds ratio of a small finger PIP joint recurrence of 4.6x. In addition to the small finger and PIP joint worsening contractures were noted to have higher recurrence rates.

The authors report the results in two models; one with surgery and subsequent injection considered as recurrence and one where repeat injection was not considered a recurrence. The same variables remained statistically significant.

Roughly 3/4 of the patients were satisfied or very satisfied.

Line 172-177.  While I agree with the authors that retreatment with injection is a significantly lesser undertaking for the patient and the surgeon, I believe that patients would consider the need for retreatment with injection a failure. The authors are to be commended for including analyses considering both repeat injection a failure and not considering repeat injection a failure. In both scenarios these data are superior to most of the studies on collagenase for DD.

Line 214-218: I would suggest that the authors speculate here on other technical aspects including perhaps the number of sites of injection and the manipulation technique as these are other variables that may result in the superior results reported here compared to most prior published studies.

Line 230: I would also add the lack of a control group as a lesser limitation. The current situation for treating DD includes three treatment options, which are all regarded as effective and appropriately safe. Most of the current controversy in DD, as the authors indicate, are the relative merits of the different treatments and the impact of these data are mitigated somewhat by the lack of a parallel comparison group.

Author Response

Thank you for your input. We have answered your comments below:  

Line 172-177.  While I agree with the authors that retreatment with injection is a significantly lesser undertaking for the patient and the surgeon, I believe that patients would consider the need for retreatment with injection a failure. The authors are to be commended for including analyses considering both repeat injection a failure and not considering repeat injection a failure. In both scenarios these data are superior to most of the studies on collagenase for DD.

  • Thank you. We agree with these comments and have added a section highlighting these issues in the Discussion (lines 234-241).

Line 214-218: I would suggest that the authors speculate here on other technical aspects including perhaps the number of sites of injection and the manipulation technique as these are other variables that may result in the superior results reported here compared to most prior published studies.

  • Thank you for the input, we have added a section about the possible role of the technical aspects in improving the results (lines 336-342).

 Line 230: I would also add the lack of a control group as a lesser limitation. The current situation for treating DD includes three treatment options, which are all regarded as effective and appropriately safe. Most of the current controversy in DD, as the authors indicate, are the relative merits of the different treatments and the impact of these data are mitigated somewhat by the lack of a parallel comparison group.

  • We have added this as a limitation and elaborated on the treatment options (lines 383-390).

Reviewer 2 Report

The authors present on longer term outcome of collagenase. They defined recurrence and looked at specific parameters that may be correlated with that. was the study ethically approved?

the parameters are unclear. Known parameters to recurrence are those linked to fibrosis diathesis and are not mentioned. the authors need to be clear at what they focus. It is unclear but are it the parameters in the tables. Now they conclude that the presence of severe PIP contractures in digit 5 is mostly prona, as it is in every Dupuytren treatment. The study needs to be presented more clearly and focussed, I believe and the conclusions and the title need to be correct: if only contractures are looked at, age sex, previous treatment and smoking (and diabetes), than that needs to be clearly described and argumented. How were these parameters selected and why? Why the subtitle DASH and satisfaction? Was this correlated as well?

Statistical review is advised. But improved study clarity and presentation is encouraged.

Author Response

Thank you for your input. We have answered your comments below:  

The authors present on longer term outcome of collagenase. They defined recurrence and looked at specific parameters that may be correlated with that. was the study ethically approved?

  • The study is ethically approved as stated in the manuscript. (line 411-412)

The parameters are unclear.Known parameters to recurrence are those linked to fibrosis diathesis and are not mentioned. the authors need to be clear at what they focus. It is unclear but are it the parameters in the tables. Now they conclude that the presence of severe PIP contractures in digit 5 is mostly prona, as it is in every Dupuytren treatment. The study needs to be presented more clearly and focussed, I believe and the conclusions and the title need to be correct: if only contractures are looked at, age sex, previous treatment and smoking (and diabetes), than that needs to be clearly described and argumented. How were these parameters selected and why? Why the subtitle DASH and satisfaction? Was this correlated as well?

Statistical review is advised. But improved study clarity and presentation is encouraged.

  • We have included information regarding the Dupuytren diathesis and potential risk factors for recurrence, such as smoking and diabetes, to the Introduction and Discussion sections (lines 28-35, 242-245 & 391-394). We have also commented on this issue in the Discussion.

  • We have also added to the statistics section information about how the covariates were selected (lines 126-129). The statistical analyses were performed by a biostatistician who is a coauthor (AÅ).

  • With regard to the subtitle “DASH and satisfaction”, we have changed the subtitle to “Patient-reported outcomes” which we believe would be more appropriate (line 172). Because patient-reported outcomes were not obtained at baseline (before treatment) it was not possible to consider them in the analysis of risk factors for recurrence. We have added a comment in the limitations part of the Discussion about this issue (lines 395-397).

Reviewer 3 Report

In general, this study conducted a good study about long-term outcomes of collagenase injection of contracture recurrence.

Introduction should include more information about the current state of collagenase injection in DD. 

I assumed that authors were indicating risk factors rather than predictor?

Author Response

Thank you for your input. We have answered your comments below:  

In general, this study conducted a good study about long-term outcomes of collagenase injection of contracture recurrence.

Introduction should include more information about the current state of collagenase injection in DD. 

  • Thank you. We have expanded the Introduction with information regarding the current state of collagenase injection in DD (lines 36-43)

I assumed that authors were indicating risk factors rather than predictor?

  • Thank you, we have changed “predictor” to “risk factor” in the text as well as in the title.